# Positive Selection of Mitochondrial *cytochrome b* Gene in the Marine Bivalve *Keenocardium buelowi* (Bivalvia, Cardiidae)

**DOI:** 10.3390/ani14192812

**Published:** 2024-09-29

**Authors:** Hyeongwoo Choi, Yeongjin Gwon, Yun Keun An, Seong-il Eyun

**Affiliations:** 1Department of Life Science, Chung-Ang University, Seoul 06974, Republic of Korea; creo9447@cau.ac.kr; 2Department of Biostatistics, University of Nebraska Medical Center, Omaha, NE 68105, USA; 3Department of Aquaculture, Chonnam National University, Yeosu 59626, Republic of Korea

**Keywords:** Cardiidae, *Keenocardium buelowi*, complete mitochondrial genome, selection pressure, phylogenetic relationship

## Abstract

**Simple Summary:**

The present study aimed to assemble and annotate the mitochondrial genome of *Keenocardium buelowi* and conducted an in-depth analysis to investigate the molecular features of this mitochondrial genome. Our findings established phylogenetic relationships within the Cardiidae family. Additionally, evidence of selection pressure on the *cytochrome b* gene was detected, suggesting its important role in the evolution of *K. buelowi*.

**Abstract:**

The mitochondrial genome provides valuable data for phylogenetic analysis and evolutionary research. In this study, we sequenced, assembled, and annotated the mitochondrial genome of *Keenocardium buelowi* using the Illumina platform. The genome spanned 16,967 bp and included 13 protein-coding genes (PCGs), two ribosomal RNAs, and 22 transfer RNAs. All PCGs utilized standard ATN start codons and TAN stop codons. The phylogenetic tree based on maximum likelihood and Bayesian inference analyses revealed Clinocardiinae as the sister group to Trachycardiinae, with the estimated divergence time being 44.5 million years ago (MYA) between *K. buelowi* and *Vasticardium flavum*. Notably, the *cytochrome b* gene (*cob*) exhibited a positive selection signal. Our findings provide valuable insights into the evolutionary history and molecular phylogeny of *K. buelowi*.

## 1. Introduction

*Keenocardium buelowi*, a marine bivalve belonging to the Cardiidae family, is found at a depth of 10–50 m and is distributed across the eastern, southern, and western seas of the Korean Peninsula [1]. In Korea, this cockle species is predominantly collected through coastal fishing. Moreover, *K. buelowi* can be distinguished by its yellow foot, in contrast to the reddish foot found in *Tegillarca granosa* [2]. As these cockles are either consumed by local residents upon capture or unrecorded by fishermen, their reported yield is very low. Therefore, comprehensive research and data on *K. buelowi* are limited. In contrast, other cockle species, such as *T. granosa* [3] and *Fulvia mutica* [4], are extensively harvested in South Korea. In 2017, approximately 115 tons of *T. granosa* were harvested in South Korea [3]. The harvest season for cockles typically extends from late autumn to early spring [5]. However, the optimal period for harvesting *K. buelowi* extends from early summer to late autumn. Recently, the documented harvested biomass of harvested *K. buelowi* has been increasing, particularly in the southern sea of Korea, highlighting their potential as a valuable fishery resource and an alternative to expensive cockles.

Despite the increasing interest in *K. buelowi*, a significant gap exists in the genomic data necessary to fully understand its biological and ecological characteristics. To address this gap, we performed a detailed analysis of the mitochondrial genome of *K. buelowi*. In animals, mitochondria exhibit several unique characteristics. In particular, they are typically maternally inherited, highly conserved, and present in multiple copies in a cell. Moreover, they have a low rate of sequence recombination and evolve rapidly [6]. Because of these characteristics, mitochondrial sequences are particularly valuable for phylogenetic research [7]. Despite the extensive application of mitochondrial genome sequences in various studies, no complete mitochondrial sequences from the Clinocardiinae subfamily are publicly available as of July 2024. Additionally, only 13 complete mitochondrial genomes from seven genera were available.

In this study, we employed high-throughput sequencing technology (next-generation sequencing) to assemble the complete mitochondrial genome of *K. buelowi*. Furthermore, we compared the mitochondrial genome of *K. buelowi* with those of other Cardiidae species to infer their phylogenetic relationships and elucidate their evolutionary histories. Our findings may enhance our understanding of the evolutionary dynamics of *K. buelowi*, providing a foundation for future research and the sustainable management of this potentially valuable fishery resource.

## 2. Materials and Methods

### 2.1. Sample Collection and Sequencing

A healthy individual sample of *K. buelowi* was collected from Yokjido Island (35°3′29′′ N, 127°44′59′′ E), South Korea, in March 2024 using a shellfish dredge. The sample was then transported to the laboratory. Subsequently, the foot tissue was dissected, preserved in 99% ethanol, and stored in a −80 °C freezer.

Whole genomic DNA was extracted using the QIAGEN Blood and Cell Culture DNA Mini Kit (QIAGEN, Hilden, Germany), according to the manufacturer’s instructions. A 150-bp paired-end library was then generated using the TruSeq DNA Nano 550 Kit (Illumina, Inc., San Diego, CA, USA) and sequenced on the NovaSeq 6000 platform (Illumina).

### 2.2. Mitochondrial Genome Assembly and Annotation

Before mitochondrial genome assembly, Trim_Galore (version 0.6.10) [8] was employed to remove reads containing low-quality bases and unknown bases (Ns) using the following options: “--quality 30 --length 121 --max_n 0” [9].

The assembly of the mitochondrial genome and annotation of protein-coding genes (PCGs) were performed using SPAdes (version 3.15.4) [10] and MitoZ (version 3.6) [11] with filtered reads. Moreover, MITOS1 [12] and MITOS2 [13] were utilized to verify the accuracy of the PCG annotation. The complete mitochondrial genome was visualized using Circos (version 0.69-8) [14].

### 2.3. Phylogenetic Analysis and Divergence Time Estimation

Two phylogenetic trees were reconstructed using datasets based on the *cytochrome c oxidase subunit I* gene (*cox1*) and 13 PCGs, respectively. For the *cox1* dataset, 23 nucleotide sequences were obtained from the National Center for Biotechnology Information (NCBI): 21 from Cardiidae species and two from Pharidae species. MAFFT (version 7.475) [15] was employed to align these 23 nucleotide sequences. Subsequently, the best-fit nucleotide model for the *cox1* dataset was identified using ModelFinder in IQ-TREE (version 2.2.0.3) [16]. For the 13 PCG dataset, only seven whole mitochondrial genomes from the Cardiidae family, including our data, were available at the NCBI (as of July 2024). Hence, nine mitochondrial genome sequences including two Pharidae species were used in this study. The nucleotide sequences of the 13 PCGs were aligned using MAFFT and concatenated. Next, PartitionFinder2 (version 2.1.1) [17] was used to determine the best-fit nucleotide model for each PCG based on the corrected Akaike information criterion (AICc).

Based on the *cox1* and 13 PCG datasets, a maximum likelihood (ML) phylogenetic tree was reconstructed using RAxML-NG (version 0.9.0). To evaluate the robustness of the ML tree, 1000 bootstrapping iterations were performed. A Bayesian inference (BI) phylogenetic tree was also reconstructed using MrBayes (version 3.2.4) [18]. When generating the BI tree, two independent Markov chain Monte Carlo (MCMC) runs consisting of 1 × 10^7^ generations were performed. To ensure convergence of the cold and heated chains, sampling was performed every 500 generations, and 25% of the samples were discarded as burn-in. FigTree (version 1.4.4) (http://tree.bio.ed.ac.uk/software/figtree, accessed on 1 July 2020) was employed to visualize the topology of the ML tree. Bootstrapping support (BS) based on the ML analysis and Bayesian posterior probability (BPP) based on the BI analysis were indicated at each node.

The divergence times of Cardiidae species were estimated using MCMCtree in PAML (version 4.10.5) [19]. The ML tree reconstructed using 13 PCGs was used as the input file. Moreover, three calibration points were utilized based on the TimeTree database [20]: *Vasticardium flavum*–*Acanthocardia tuberculata* (100.9–214.9 million years ago [MYA]), *Hippopus porcellanus*–*Tridacna derasa* (51.6 MYA), and the root age (210.1–514.5 MYA).

### 2.4. Positive Selection Analysis

To elucidate genomic response potentially associated with adaptation to environmental pressures, the CodeML program within the PAML package [19] was utilized to identify positively selected genes (PSGs) among the mitochondrial PCGs. The ML tree constructed using 13 mitochondrial genomes was utilized; the genomes were unrooted, and their branch lengths were removed (Figure 4a). Then, PAL2NAL (version 14) [21] was employed to align the nucleotide sequences.

A branch-site model was used to perform selection analysis. Using the branch-site model, a null model (model = 2, NSsites = 2, fix_omega = 1) was compared with an alternative model (model = 2, NSsites = 2, fix_omega = 0). This enabled the ratio of nonsynonymous (dN) to synonymous (dS) substitutions to vary among codon sites and the dN/dS ratio in the foreground branch to differ from that in the background branches. Likelihood ratio tests (LRTs) and chi-square (χ2) distributions were employed to determine the best-fit models and assess the statistical significance of the data (*p* values). If the LRT denoted statistical significance (*p* < 0.05) and the ω (dN/dS) ratio was greater than 1, the site was considered to be under positive selection.

AlphaFold2 (version 2.3.1) [22] was used to predict the three-dimensional (3D) structures of the 13 PCGs. PyMol (version 4.6) was employed for visualization [23].

## 3. Results and Discussions

### 3.1. General Information Regarding the Mitochondrial Genome of K. buelowi

In this study, 765,609,380 raw reads, equivalent to 116 Gb, were generated from the *Keenocardium buelowi* genome using Illumina paired-end sequencing. By following a stringent quality control process (Phred quality > 30), 653,573,610 clean reads were obtained, accounting for 85.37% of the original dataset (Table 1). These high-quality reads were then used to assemble the mitochondrial genome of *K. buelowi*.

The complete mitochondrial genome of *K. buelowi* spanned 16,967 bp (Figure 1; GenBank: PP760258.1). The genome included 13 PCGs (*atp6*, *atp8*, *cox1*, *cox2*, *cox3*, *cob*, *nad1*, *nad2*, *nad3*, *nad4*, *nad4l*, *nad5*, and *nad6*), two ribosomal RNA (rRNA) genes (*rrnS* and *rrnL*), and 22 transfer RNA (tRNA) genes (*trnY*, *trnW*, *trnK*, *trnP*, *trnV*, *trnM*, *trnR*, *trnQ*, *trnL2*, *trnL1*, *trnF*, *trnT*, *trnE*, *trnS2*, *trnS1*, *trnH*, *trnG*, *trnC*, *trnN*, *trnA*, *trnI*, and *trnD*) (Table 2). Thirty-seven genes were located on the heavy strand and accounted for 88.25% of the entire mitochondrial genome. The 13 PCGs accounted for 64.71% of the entire mitochondrial genome. Most PCGs in *K. buelowi* began with either of the two start codons ATA (eight genes) or ATG (five genes). The stop codon employed in the mitochondrial genome was either TAG (seven genes) or TAA (six genes) (Table 2).

### 3.2. Results of Phylogenetic Analysis

Although the construction of phylogenetic trees using multiple genes across various species is typically more informative than that using a single gene, we initially aimed to use 13 mitochondrial PCGs to elucidate phylogenetic relationships. However, due to the lack of complete mitochondrial genome sequences within the Clinocardiinae subfamily, *cox1* sequences from 23 species were used instead (Figure 2). Moreover, a phylogenetic tree was reconstructed using 13 mitochondrial PCGs from nine species (Figure 3). Our study is important as it is the first to unravel the mitochondrial sequence within the Clinocardiinae subfamily, shedding light on taxonomic relationships at the subfamily level.

ML and BI trees were constructed using two datasets: *cox1* and 13 mitochondrial PCG datasets. For the *cox1*, the best-fit model was GTR + F + I + G4 according to the AICc. Since the *atp8* gene was unavailable in three species (*Acanthocardia tuberculata*, *Vasticardium flavum*, and *Sinnonovacula constricta*), we manually reannotated it. For the 13 concatenated PCGs dataset, GTR + I + G was used for four genes (*cox1*, *nad2*, *nad4l*, and *nad6*), GTR + G was used for *cox2*, TVM + G was used for three genes (*atp6*, *atp8*, and *cox3*), and TVM + I + G was used for five genes (*cob*, *nad1*, *nad3*, *nad4*, and *nad5*). The ML and BI trees based on 13 PCGs showed identical topologies. From our study based on the *cox1* dataset, Trachycardiinae was found to be the sister group to Clinocardiinae (BS = 77, BPP = 1) (Figure 2), consistent with the results based on the 13 PCG dataset (BS = 92, BPP = 1) (Figure 3). However, the relationships between Tridacninae and Lymnocardiinae differed in the *cox1* and 13 PCG datasets (Figure 2 and Figure 3). The *cox1* dataset exhibited weak node confidence (BS ≤ 32, BPP ≤ 0.5), whereas the 13 PCG dataset exhibited high node support (BS ≥ 90, BPP = 1) (Figure 2 and Figure 3). As trees based on multiple genes (13 genes in this study) generally provide stronger phylogenetic signals, we propose that Lymnocardiinae is the sister group to the Clinocardiinae + Trachycardiinae + Cardiinae clade. However, no complete mitochondrial data are currently available for Clinocardiinae. Moreover, only six complete mitochondrial sequences exist for Cardiidae. To better understand the phylogenetic positions of Cardiidae, additional mitochondrial genome sequences will be required.

The divergence time between *K. buelowi* and *V. flavum* was estimated to be 44.5 MYA (95% CI: 18–60.9) (Figure 3). Moreover, the divergence time between *Cerastoderma edule* and *A. tuberculata* was estimated to be 43.9 MYA (95% CI: 17.8–60.2), consistent with the findings of Li et al. [24]. The divergence time between *V. flavum* and *F. mutica* overlapped with the estimates provided by Herrera et al. [25]. However, the rapid substitution rates in the mitochondrial genome hinder the precise estimation of deep divergence times [26]. Given the influence of horizontal gene transfer, frequent introgression, and incomplete lineage sorting, phylogenetic research solely based on mitochondrial sequences is inherently limited [27,28]. Therefore, nuclear and mitochondrial sequences may provide contrasting or complementary information regarding tree topologies and branch lengths [29,30].

### 3.3. Selection Patterns in Mitochondrial Genes of K. buelowi

Mitochondrial genes play a vital role in cellular function, and as a result, purifying selection is the primary force shaping their evolution. Therefore, identifying signatures of positive selection provide valuable insights into how these genes may contribute to adaptive responses to the environment pressures. To identify genes subjected to environmental pressures, the ω ratio (dN/dS) was calculated for each of the 13 mitochondrial PCGs and compared (Figure 4 and Table 2). The ω ratio was used to determine the natural selection patterns for these PCGs [31]. The branch-site model revealed that all PCGs, except for *cob*, exhibited signals of purifying selection during evolution (Table 3). However, the results of *cob* were significant in the LRT (LR = 6.97, *p* < 0.008), suggesting that it underwent strong selection pressure throughout its evolutionary history. Specifically, residue 256 in six Cardiidae species was isoleucine, a hydrophobic amino acid, which was substituted by aspartic acid, a negatively charged amino acid, in *K. buelowi.* This alteration suggests a potential adaptive response. However, the functional properties of these substitutions still need to be elucidated through further physiological experiments. Jacobsen et al. reported that mutations in mitochondrial genes are linked to their positions within the mitochondrial genome [32]. This aligns with our findings that *cob* may have evolved more rapidly because of being distant from the origin of mitochondrial genome replication (Figure 1 and Table 3) [33]. As protein structures and functions depend on amino acid interactions, the placement of Asp256 within the 3D structure was further examined in our study (Figure 4b). The positively selected residue was located close to the periplasmic membrane region (Figure 4c). Moreover, *cob* exhibited strong positive selection pressure, suggesting that *cob* plays a crucial role in the evolution of the *K. buelowi* mitochondrial genome. However, the correlation between positive selection pressure on *cob* and environmental variables was limited in our study. Nonetheless, this result may be influenced by other variables not evaluated in our study.

## 4. Conclusions

In this study, next-generation sequencing was performed to assemble and annotate the 16,967-bp mitochondrial genome of *Keenocardium buelowi*, providing the first complete mitochondrial genome within the Clinocardiinae subfamily. Our comprehensive analysis assessed the molecular characteristics of the *K. buelowi* mitochondrial genome. ML and BI trees were reconstructed based on *cox1* and 13 PCG datasets. Our findings indicated that Clinocardiinae forms a monophyletic group comprising two genera: *Keenocardium* and *Clinocardium*. Moreover, our data confirmed that Clinocardiinae is the sister group to Trachycardiinae. The divergence times for seven Cardiidae species were estimated using 13 mitochondrial PCGs. Furthermore, our results revealed that *cob* shows signs of selection pressure, suggesting its significant role in the evolution of *K. buelowi*. Further research is needed to improve the resolution of phylogenetic relationships among Cardiidae species since complete mitochondrial genome for this family is still lacking.

## Figures and Tables

**Figure 1 animals-14-02812-f001:**
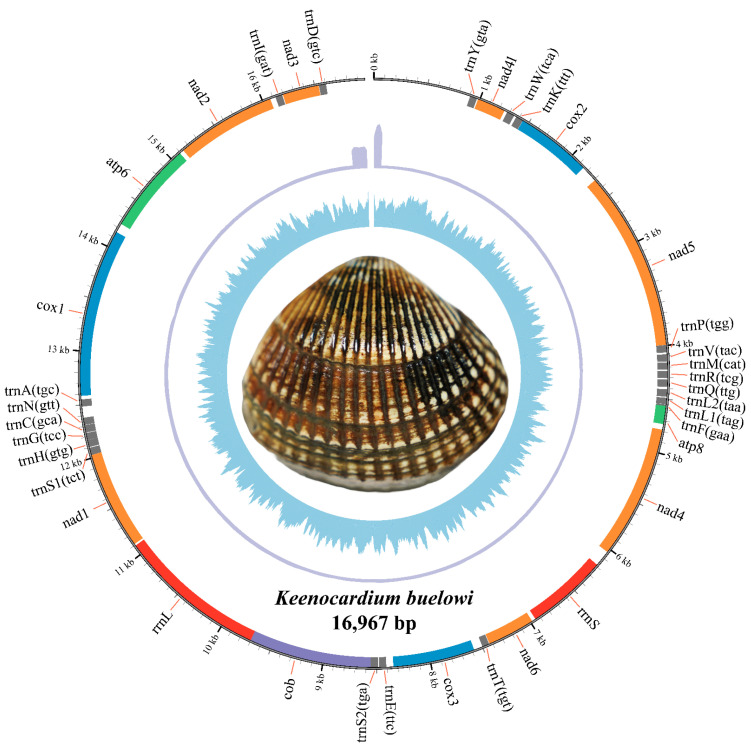
The complete mitochondrial genome circular map of *Keenocardium buelowi*. The inner circle represents GC contents, the middle circle indicates sequencing depth, and the outer circle represents gene arrangements. Grey, orange, blue, green, red, and purple colors refer to *trna*, *nad*, *cox*, *atp*, *rrna*, and *cob*, respectively.

**Figure 2 animals-14-02812-f002:**
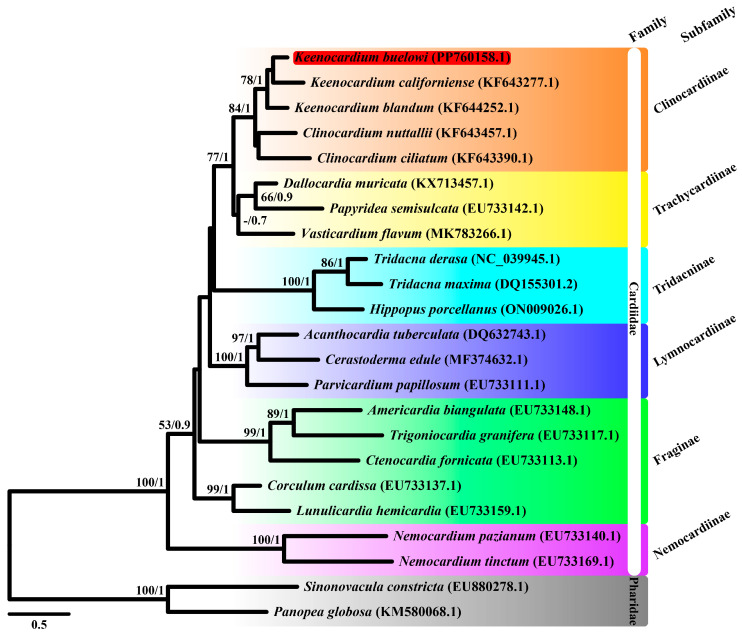
Maximum likelihood (ML) tree of 21 Cardiidae species and two Pharidae species based on mitochondrial *cytochrome c oxidase subunit I* gene (*cox1*) sequences. Sequences from *Sinonovacula constricta* (EU880278.1) and *Panopea globosa* (KM580068.1) were used as outgroup species. *Keenocardium buelowi* was highlighted in red. The numbers on each node represent the bootstrap support (BS) from ML analysis and Bayesian posterior probability (BPP) from Bayesian inference analysis. Node supports with BS ≥ 60 and BPP ≥ 0.7 are indicated on each node. The scale bar represents the number of substitutions per site.

**Figure 3 animals-14-02812-f003:**
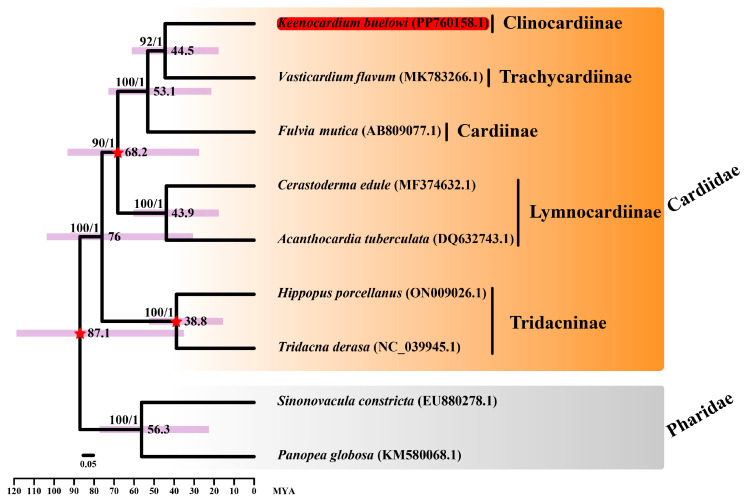
Divergence time estimation analysis of seven Cardiidae species and two Pharidae species. Sequences from *Sinonovacula constricta* (EU880278.1) and *Panopea globosa* (KM580068.1) were used as outgroup species. *Keenocardium buelowi* was highlighted in red. The three calibration times used in this study are denoted with a red star in each node. The number on each node indicates the bootstrap support (BS) from ML analysis and Bayesian posterior probability (BPP) from Bayesian inference analysis. Node supports with BS ≥ 60 and BPP ≥ 0.7 are indicated on each node. Purple bars represent 95% highest posterior density (HPD) intervals, and the scale bar represents the number of substitutions per site.

**Figure 4 animals-14-02812-f004:**
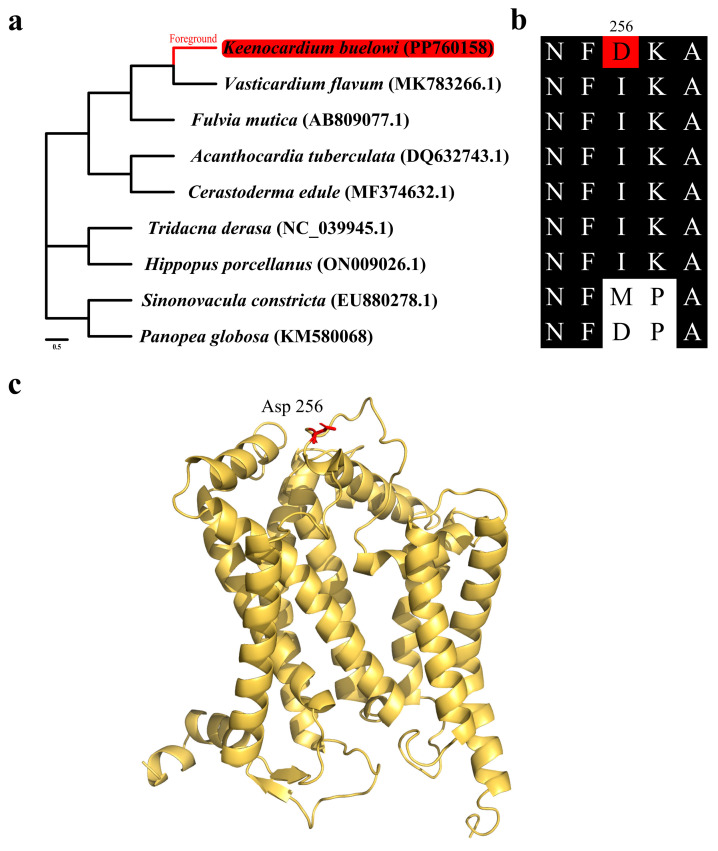
Positive selection analysis results of the mitochondrial *cytochrome b* gene (*cob*). (**a**) Species tree used for positive selection analysis. The *Keenocardium buelowi* branch was used as a foreground. (**b**) Multiple sequence alignment (MSA) of mitochondrial *cob* from nine bivalves. Within the alignment, the red background represents mutated residues (Asp256), while the black background represents highly conserved regions. (**c**) Yellow ribbon diagram showing the three-dimensional structure of mitochondrial *cob*. The red residue (Asp256) indicates a positively selected site from the branch-site model.

**Table 1 animals-14-02812-t001:** General information on next-generation sequencing and mitochondrial genome assembly.

	*Keenocardium buelowi*	
Sequencing	Platform	Illumina NovaSeq 6000
	Library kit	TruSeq DNA Nano
	Read length (bp)	151 × 2
	Insert size (bp)	550
	Number of reads	765,609,380
	Mean quality score	34.69
	% of reads ≥ Q30 (%)	86.54
	Number of bases (bp)	115,607,016,380 (116 Gb)
Data filtering	Number of reads	653,573,610
	Number of bases (bp)	97,914,308,918 (98 Gb)
Mitochondrial genome assembly	Total length (bp)	16,967
	GC content (%)	39.07
	Number of protein-coding genes	13

**Table 2 animals-14-02812-t002:** Organization of the *Keenocardium buelowi* mitochondrial genome.

Gene	Position	Length (bp)	Start Codon	Stop Codon	Strand
*trnY*	906–973	68			+
*nad4l*	984–1250	267	ATA	TAA	+
*trnW*	1278–1350	73			+
*trnK*	1373–1442	70			+
*cox2*	1444–2154	711	ATG	TAA	+
*nad5*	2310–3995	1686	ATG	TAG	+
*trnP*	3996–4065	70			+
*trnV*	4082–4149	68			+
*trnM*	4158–4224	67			+
*trnR*	4236–4304	69			+
*trnQ*	4317–4385	69			+
*trnL2*	4407–4476	70			+
*trnL1*	4486–4545	60			+
*trnF*	4547–4616	70			+
*atp8*	4651–4731	81	ATA	TAA	+
*nad4*	4784–6034	1251	ATG	TAG	+
*rrnS*	6185–6946	762			+
*nad6*	6982–7428	447	ATG	TAG	+
*trnT*	7433–7499	67			+
*cox3*	7581–8345	765	ATA	TAG	+
*trnE*	8411–8478	68			+
*trnS2*	8488–8555	68			+
*cob*	8556–9701	1146	ATA	TAG	+
*rrnL*	9701–11,093	1393			+
*nad1*	11,110–12,027	918	ATA	TAA	+
*trnS1*	12,028–12,097	70			+
*trnH*	12,104–12,170	67			+
*trnG*	12,171–12,237	67			+
*trnC*	12,242–12,311	70			+
*trnN*	12,314–12,380	67			+
*trnA*	12,484–12,551	68			+
*cox1*	12,584–14,155	1572	ATG	TAG	+
*atp6*	14,241–15,095	855	ATA	TAA	+
*nad2*	15,135–16,073	939	ATA	TAG	+
*trnI*	16,124–16,189	66			+
*nad3*	16,196–16,537	342	ATA	TAA	+
*trnD*	16,541–16,607	67			+

**Table 3 animals-14-02812-t003:** Results of positive selection analysis using branch-site model. Abbreviations: np (number of parameters), LRT (likelihood ratio test), BEB (Bayes empirical Bayes).

Gene Name	Null Model (np)	Alternative Model (np)	LRTs (*p*-Value)	Site Class	0	1	2a	2b	Positively Selected Site (BEB)
*atp6*	−2917.107385 (19)	−2917.107385 (20)	0 (1)	proportion	0.9168	0.04076	0.04064	0.00181	
				background ω	0.01034	1	0.01034	1	
				foreground ω	0.01034	1	1	1	
*atp8*	−431.861892 (19)	−431.861892 (20)	0 (1)	proportion	0.68114	0.31886	0	0	
				background ω	0.01507	1	0.01507	1	
				foreground ω	0.01507	1	1	1	
*cox1*	−6547.79 (19)	−65,473.56 (20)	0.45 (0.5)	proportion	0.98139	0.01061	0.00791	0.00009	
				background ω	0.0074	1	0.0074	1	
				foreground ω	0.0074	1	97.76176	97.76176	
*cox2*	−3707.05 (19)	−3708.08 (20)	2.072 (0.15)	proportion	0.83713	0.16287	0	0	
				background ω	0.02415	1	0.02415	1	
				foreground ω	0.02415	1	1	1	
*cox3*	−3514.15 (19)	−3514.15 (20)	0 (1)	proportion	0.89204	0.10796	0	0	
				background ω	0.01727	1	0.01727	1	
				foreground ω	0.01727	1	1	1	
*cob*	−5259.88 (19)	−5263.37 (20)	6.97 (0.008)	proportion	0.94301	0.05699	0	0	256D (0.545)
				background ω	0.01445	1	0.01445	1	
				foreground ω	0.01445	1	2.88836	2.88836	
*nad1*	−3993.63 (19)	−3993.63 (20)	0 (1)	proportion	0.92734	0.07266	0	0	
				background ω	0.01206	1	0.01206	1	
				foreground ω	0.01206	1	1	1	
*nad2*	−3946.12 (19)	−3946.12 (20)	0 (1)	proportion	0.60777	0.17795	0.16575	0.04853	
				background ω	0.04311	1	0.04311	1	
				foreground ω	0.04311	1	1	1	
*nad3*	−1609.36 (19)	−1609.06 (20)	0.6 (0.44)	proportion	0.75983	0.0924	0.13175	0.01602	
				background ω	0.00991	1	0.00991	1	
				foreground ω	0.00991	1	61.37822	61.37822	
*nad4*	−4285.57 (19)	−4285.47 (20)	0.21 (0.65)	proportion	0.86858	0.098	0.03003	0.00339	
				background ω	0.02398	1	0.02398	1	
				foreground ω	0.02398	1	18.93539	18.93539	
*nad4l*	−1052.15 (19)	−1052.15 (20)	0 (1)	proportion	0.77777	0.22223	0	0	
				background ω	0.04722	1	0.04722	1	
				foreground ω	0.04722	1	1	1	
*nad5*	−6938.05 (19)	−6938.05 (20)	0 (1)	proportion	0.77041	0.22959	0	0	
				background ω	0.04166	1	0.04166	1	
				foreground ω	0.04166	1	1	1	
*nad6*	−1651.78 (19)	−1652.6 (20)	1.64 (0.2)	proportion	0.73773	0.26227	0	0	
				background ω	0.05979	1	0.05979	1	
				foreground ω	0.05979	1	1	1	

## Data Availability

The complete mitochondrial genome sequence of *Keenocardium buelowi* is available in GenBank under the accession number PP760158.1.

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
