# Peer review of "Positive Selection of Mitochondrial cytochrome b Gene in the Marine Bivalve Keenocardium buelowi (Bivalvia, Cardiidae)"

_animals, 2024, doi:10.3390/ani14192812_

Round 1

Reviewer 1 Report

Comments and Suggestions for Authors

Please see my comments in the attached PDF file.

Comments on the Quality of English Language

The quality of the English language is good. I have only a few minor comments to improve clarity here and there.

Author Response

Review of “Positive selection of mitochondrial cytochrome b gene in the marine bivalve Keenocardium buelowi (Bivalvia, Cardiidae)” by Choi et al. Submitted to the journal Animals.
I have some minor editorial comments itemized below. I only have one significant suggestion for additional analysis of mitochondrial genome annotation in close relatives of K. buelowi. This additional analysis may allow the authors to conduct further tests if they are able to annotate atp8 in some additional bivalve genomes.

Comments 1: l. 10. …annotate the mitochondrial genome…

Response 1: As suggested by our reviewer, we have revised accordingly in line 10.

Comments 2: l. 11. …conducted an in- depth analysis…

Response 2: As suggested by our reviewer, we have revised accordingly in line 11.

Comments 3: l. 15. Replace essential with valuable.

Response 3: As suggested by our reviewer, we have revised accordingly in line 15.

Comments 4: ll. 33-34. I do not know what you mean by “self-consumed locally.”Clarify.

Response 4: As suggested by our reviewer, we have revised accordingly in lines 34-35:

          “As these cockles are either consumed by local residents upon capture…”

Comments 5: l. 36. By “extensively used”, do you mean something like “extensively harvested and sold commercially, consequenctly considerably more data are available for these species than for K. buelowi.”

Response 5: As suggested by our reviewer, we have revised and add harvest data in lines 37-39. Please note that due to the limited availibitility of harvest data for F. mutica, we have included information on T. granosa instead.

          “…are extensively harvested…”

          “In 2017, approximately 115 tons of T. granosa were harvested in South Korea

Comments 6: ll. 36-38. DO you mean “…the harvesting season for T. granosa and F. mutica typically occurs during the cooler months”?

Response 6: We have clarified the sentence in line 39.

Comments 7: l. 39. …the documented harvested biomass…

Response 7: We have revised according to our reviewer’s suggestion.

Comments 8: l. 45 Start the sentence with “In animals, mitochondria exhibit several…”

Response 8: We have revised according to our reviewer’s suggestion in line 47.

Comments 9: l. 45. “…they are typically maternally inherited…”

Response 9: We have revised according to our reviewer’s suggestion in line 48.

Comments 10: l. 66. “Whole genomic DNA…”

Response 10: We have revised according to our reviewer’s suggestion in line 70.

Comments 11: l. 81. At the end of the sentence add, “…13 PCGs, respectively.”

Response 11: We have revised according to our reviewer’s suggestion in line 87.

Comments 12: l. 82. Put cox1 in italics here and throughout.

Response 12: As suggested by our reviewer, we have italicized throughout the manuscript.

Comments 13: L. 113. I suspect that the atp8 gene is actually present in the additional three genomes. As noted by Breton et al. 2010, atp8 evolves relatively rapidly compared to other mtDNA genes and may be missed in standard mitochondrial genome annotations. The authors of the current paper may wish to examine the three genomes where the atp8 gene is thought to be missing. In particular, they may wish to examine the region of the genome where atp8 would be expected to occur based on an alignment of closely related taxa for which atp8 has been identified. IS there sequence there? Can you identify possible start and stop codons? If so, you may consider reanalysing the data using any additional atp8 data that you can find. If you do identify missing atp8 genes in any additional genomes, this is also useful information. For example, a very quick examination shows that there is unannotated sequence between trnF and ND4 in V. flavum.

A very quick online translation of this produced this amino acid sequence, MYISVSVAVFLFSVVIWWGGKRKYNF, which matches atp8 of Ceastoderma
edule, so it is worth further investigation.
ATP synthase F0 subunit 8 [Cerastoderma edule]
Sequence ID: YP_009420862.1Length: 37Number of Matches: 1
• See 1 more title(s) See all Identical Proteins(IPG)
Related Information
Gene-associated gene details
AlphaFold Structure-3D structure displays
Identical Proteins-Identical proteins to YP_009420862.1
Range 1: 12 to 37GenPeptGraphics Next Match Previous Match
Alignment statistics for match #1
Score Expect Identities Positives Gaps
68.5 bits(154) 9e-15 20/26(77%) 21/26(80%) 0/26(0%)
Query 1 MYISVSVAVFLFSVVIWWGGKRKYNF 26
 MY+SVS VFLF VVIWW GKR YNF
Sbjct 12 MYVSVSLSVFLFMVVIWWSGKRAYNF 37
(Reference: Breton, S., Stewart, D. T., & Hoeh, W. R. (2010). Characterization of a
mitochondrial ORF from the gender-associated mtDNAs of Mytilus spp.(Bivalvia: Mytilidae): identification of the “missing” ATPase 8 gene. Marine Genomics, 3(1), 11-18.)

Response 13: Thank you for your suggestion. We have reannotated atp8 from other three species manually. And we updated Figure 3, Table 3 and sentence in lines 179-184.

Comments 14: l. 111. The meaning of the phrase “genomic adaptation mechanisms” is unclear to me in this context. Clarify.

Response 14: We have clarified the sentence in lines 118-119 as follows:

          “To elucidate genomic response potentially associated with adaptation to environmental pressures,…”

The remaining phylogenetic analyses and analysis of selection in the genome are appropriately conducted and interesting. They provide some useful information on an understudied taxon.

We appreciate the reviewer for their thoughtful insights and suggestions.

Reviewer 2 Report

Comments and Suggestions for Authors

The biomass of harvested Keenocardium buelowi has been increasing, particularly in the southern sea of Korea, highlighting their potential as a valuable fishery. The present study assembled and annotated mitochondrial genome of K. buelowi, and detected. selection pressure on the cytochrome b gene. The study provides valuable insights into the evolutionary history and molecular phylogeny of K. buelowi. Before accept, minor revision is needed.

Minor comments:

1.      Add the study progress on complete mitochondrial sequences of Cardiidae family in introduction.

2.      In 3.1. Employing high-throughput sequencing technology to assemble the complete mitochondrial genome is nice method, and the cost is related to the sequence depth, Pls give some good advice for sequecing depth.

3.      clerical error , 5. Conclusions, the number is 4.

Comments on the Quality of English Language

The English is good.

Author Response

The biomass of harvested Keenocardium buelowi has been increasing, particularly in the southern sea of Korea, highlighting their potential as a valuable fishery. The present study assembled and annotated mitochondrial genome of K. buelowi, and detected. selection pressure on the cytochrome b gene. The study provides valuable insights into the evolutionary history and molecular phylogeny of K. buelowi. Before accept, minor revision is needed.

Minor comments:

Comments 1: Add the study progress on complete mitochondrial sequences of Cardiidae family in introduction.

Response 1: According to our reviewer, we have included related information in lines 54-55 .

Comments 2: In 3.1. Employing high-throughput sequencing technology to assemble the complete mitochondrial genome is nice method, and the cost is related to the sequence depth, Pls give some good advice for sequecing depth.

Response 2: Thank you for the suggestion. However, to refer to sequencing depth, calculating the genome size is required, which is beyond the scope of this study. Alternatively, the sequencing cost can be inferred from the generated sequencing data, which is 116 Gb in this study.

Comments 3: clerical error, 5. Conclusions, the number is 4.

Response 3: We have revised it according to our reviewer.

Reviewer 3 Report

Comments and Suggestions for Authors

This manuscript reports the complete mitochondrial genome sequence of the bivalve Keenocardium buelowi. The authors first use the data on cytochrome oxidase I to estimate the phylogyeny of 21 Cardiidae species, then use the data on 13 mitochondrial protein coding genes to estimate the phylogeny of 7 Cardiidae species. Herrera et al. (2015) used sequences of three genes to estimate a phylogeny for 110 species of Cardiidae, but the authors of this manuscript only briefly mention Herrera et al. (2015) and do not compare the topology of their tree to the tree in Herrera et al. There are numerous papers on Cardiidae systematics that are cited in Herrera et al. but ignored in this manuscript. I am not a cockle systematics expert, so it is completely unclear to me whether the phylogentic conclusions in this manuscript contradict previous research or confirm it. The authors need to expand their Introduction and Discussion to put their results in the context of earlier research.
     The second conclusion is that there is positive selection on cytochrome oxidase b. The authors repeatedly call this "strong selective pressure," but iheir conclusion is based on a single amino acid substitution (isoleucine to aspartic acid at site 256 in Keenocardium buelowi). Synonymous sites in mitochondrial genes evolve quickly and, based on figure 2, must be nearly saturated with variation when comparing different subfamilies within the Cardiidae.   I haven't looked for it, but I imagine that there is extensive literature on the effects of synonymous site saturation on the statistical tests for selection that they use. The authors need to discuss this topic. If they remain convinced that the substitution at site 256 is due to positive selection, they should compare it to other examples of positive selection in mitochondrial genes, and at cytocchrome oxidase b in particular. Is this the first example of positive selection on a mitochondrial gene, or is it a common result? Based on this manuscript, I have no idea.

Author Response

Comments 1: This manuscript reports the complete mitochondrial genome sequence of the bivalve Keenocardium buelowi. The authors first use the data on cytochrome oxidase I to estimate the phylogyeny of 21 Cardiidae species, then use the data on 13 mitochondrial protein coding genes to estimate the phylogeny of 7 Cardiidae species. Herrera et al. (2015) used sequences of three genes to estimate a phylogeny for 110 species of Cardiidae, but the authors of this manuscript only briefly mention Herrera et al. (2015) and do not compare the topology of their tree to the tree in Herrera et al. There are numerous papers on Cardiidae systematics that are cited in Herrera et al. but ignored in this manuscript. I am not a cockle systematics expert, so it is completely unclear to me whether the phylogentic conclusions in this manuscript contradict previous research or confirm it. The authors need to expand their Introduction and Discussion to put their results in the context of earlier research.
Response 1: We attempted to compare our results with those of Herrera et al. (2015) to confirm the findings, but we identified some inaccuracies in taxonomic classification. For instance, the genus Fulvia is classified under Cardiinae (as per Marine Species Database), but in Figure 2, they are included in Laevicardiinae. Additionally, several subfamilies exhibited polyphyletic topologies. However this might be due to taxonomic reclassifications. For example, Laevicardium biradiatum is currently recognized as Acrosterigma biradiatum.

    As highlighted above, there are limitations to directly referencing the results from Herrera et al. (2015).

Comments 2: The second conclusion is that there is positive selection on cytochrome oxidase b. The authors repeatedly call this "strong selective pressure," but their conclusion is based on a single amino acid substitution (isoleucine to aspartic acid at site 256 in Keenocardium buelowi). Synonymous sites in mitochondrial genes evolve quickly and, based on figure 2, must be nearly saturated with variation when comparing different subfamilies within the Cardiidae.   I haven't looked for it, but I imagine that there is extensive literature on the effects of synonymous site saturation on the statistical tests for selection that they use. The authors need to discuss this topic. If they remain convinced that the substitution at site 256 is due to positive selection, they should compare it to other examples of positive selection in mitochondrial genes, and at cytocchrome oxidase b in particular. Is this the first example of positive selection on a mitochondrial gene, or is it a common result? Based on this manuscript, I have no idea.

Response 2: Mitochondrial genes play a vital role in cellular function, and as a result, purifying selection is the primary force shaping their evolution. Therefore, identifying signatures of positive selection provide valuable insights into how these genes may contribute to adaptive responses to the environment pressures To this end, several studies have employed amino acid substitution analyses using the branch-site model to detect positive selection. Our analysis identified significant positive selection on the cob gene (p < 0.008). Specifically, residue 256 in six Cardiidae species was isoleucine, a hydrophobic amino acid, which was substituted by aspartic acid, a negatively charged amino acid, in K. buelowi. This alteration suggests a potential adaptive response. However, the functional properties of these substitutions still need to be elucidated through further physiological experiments. 

    We have added this information in lines 230-233 and 240-244.

Round 2

Reviewer 1 Report

Comments and Suggestions for Authors

There are a couple of grammatical or typographical errors as follows:

l. 55. The plural of “genus” is “genera”.

l. 70. Replace “genomice” with “genomic”.

l. 242. Put “K. buelowi”  in italic font (K. buelowi).

Comments on the Quality of English Language

It is now acceptable.

Reviewer 2 Report

Comments and Suggestions for Authors

The manuscript have riveses the commets, now it may be accepted.